# Improved Electrical Performance of InAlN/GaN High Electron Mobility Transistors with Post Bis(trifluoromethane) Sulfonamide Treatment

**Siheng Chen** [1], **Peng Cui** [1,*], **Mingsheng Xu** [1], **Zhaojun Lin** [1], **Xiangang Xu** [1], **Yuping Zeng** [2,*] and **Jisheng Han** [1,*]

[1]  Institute of Novel Semiconductors, School of Microelectronics, Shandong University, Jinan 250100, China
[2]  Department of Electrical and Computer Engineering, University of Delaware, Newark, DE 19716, USA
*  Correspondence: pcui@sdu.edu.cn (P.C.); yzeng@udel.edu (Y.Z.); j.han@sdu.edu.cn (J.H.)

**Abstract:** An enhancement of the electrical performance of the InAlN/GaN high electron mobility transistors (HEMTs) is demonstrated by the incorporation of post bis(trifluoromethane) sulfonamide (TFSI) treatment. The surface treatment of TFSI solution results in the increase of 2DEG electron mobility from 1180 to 1500 $cm^2/Vs$ and thus a reduction of on-state resistance and an increase in transconductance. The results indicate that the positive charge of $H^+$ will decrease the polarization charges of the InAlN barrier under the access region due to the converse piezoelectric effect, leading to the reduced polarization Coulomb field (PCF) scattering in InAlN/GaN HEMT. This offers a possible way to improve the electron mobility and device performance of InAlN/GaN HEMTs for further application.

**Keywords:** InAlN/GaN HEMT; electrical performance; bis(trifluoromethane) sulfonamide; electron mobility; scattering

## 1. Introduction

Recently, gallium nitride (GaN) has attracted extensive attention worldwide and become a hot topic for researchers and industries [1–5]. As an important GaN material system, lattice-matched InAlN/GaN HEMTs have attracted much attention, and it is hoped that they can replace the conventional AlGaN/GaN HEMT in certain application fields [6–10]. The InAlN can be grown on a lattice-matched GaN buffer layer with an In content of 17%, which can effectively avoid the material degradation with the strain relaxation from the lattice mismatch [11–13]. Another important reason is that the spontaneous polarization of the InAlN/GaN structure is significantly higher than the total polarization of traditional AlGaN/GaN heterostructures [14,15], resulting in a higher two-dimensional electron gas (2DEG) in the InAlN/GaN heterostructure. Compared with Si, GaAs, etc., the InAlN/GaN heterostructure possesses distinct advantages in terms of material parameters, such as band gap, critical breakdown electric field, and electron saturation drift velocity [16,17]. The device performance (including, for example, lower on-resistance, the high electron density, and the high breakdown field) clears the way for a massive adoption of InAlN/GaN HEMTs in the field of high-temperature, high-frequency microwave power devices, as well as high-voltage and low-loss power electronic devices [18,19].

However, in comparison with AlGaN/GaN heterostructures, InAlN/GaN heterostructures feature lower electron mobility, which degrades the electrical performance of InAlN/GaN HEMTs and thus limits their device application. In order to improve the electron mobility and device performance, some surface treatment methods, such as $O_2$ plasma, KOH, and some acid solution surface treatment techniques, have been investigated [20–26]. Lee et al. applied an oxygen plasma treatment on the $In_{0.17}Al_{0.83}N$/GaN HEMTs. Due to the formation of a thin oxide layer on the material surface, the reduced gate leakage current with two orders of magnitude, the suppressed transconductance collapse, and a

high current gain cutoff frequency ($f_T$) of 245 GHz were demonstrated [20]. With KOH surface treatment, Ganguly et al. reported that InAlN/GaN HEMTs exhibited a substantial reduction in the gate leakage by ~3 orders, a lower subthreshold slope (SS) by ~100 mV/dec, and improved breakdown characteristics [22]. Bis(trifluoromethane) sulfonamide (TFSI), as an organic superacid, has been used for surface treatment to improve the performance of devices [25–27]. Lin et al. reported that TFSI surface treatment improved effective field effect electron mobility ($\mu_{eff}$) by ~4.5 fold and reduced the SS by ~0.86 folds in a $MoS_2$ transistor, demonstrating a possible way to improve the device performance of $MoS_2$ transistors [25]. Zeng et al. investigated the effects of post-TFSI surface treatment on InAs FinFETs and found a prominent reduction in sheet resistance and an increase in carrier mobility to 1378 $cm^2$/Vs, a ~7.1-fold enhancement, with the TFSI treatment [26]. Based on the previous reports, the improved electron mobility on $MoS_2$ and InAs transistors was respectively demonstrated with the TFSI surface treatment. However, to the best of our knowledge, there have been no previous studies on TFSI treatment on GaN HEMTs and the influences are not clear.

## 2. Results

In this paper, the influence of TFSI surface treatment on InAlN/GaN HEMTs is investigated. After immersing the InAlN/GaN HEMT in the TFSI solution for a certain time, the $H^+$ in the solution will neutralize the polarization charge in the InAlN barrier layer and then reduce the polarization Coulomb field (PCF) scattering. With post-TFSI surface treatment, the 2DEG electron mobility of InAlN/GaN HEMTs reached 1500 $cm^2$/Vs, which was about 30% higher than the device before TFSI treatment. Meanwhile, the overall performances of the devices were effectively improved due to the reduction of on-state resistance and the increase in transconductance.

Figure 1 shows the schematic of the device fabrication process. As shown in Figure 1a, a Si substrate was used and the epilayer was grown by metalorganic chemical vapor deposition (MOCVD). The thickness of the GaN buffer layer was 2 μm, and the thickness of the InGaN back barrier with an In content of 12% was 4 nm. Then, a 15-nm GaN channel layer, 1-nm AlN interlayer, 8-nm $In_{0.17}Al_{0.83}N$ barrier, and 2-nm GaN cap layer were grown to form an InAlN/GaN heterostructure. Hall measurements at room temperature showed that the electron concentration and mobility were $2.28 \times 10^{13}$ $cm^{-2}$ and 1205 $cm^2$/V·s, respectively. Figure 1b shows that mesa isolation was carried out with the inductively coupled plasma (ICP) etching. A $Cl_2$/$BCl_3$ gas mixture was used, and the etching depth was ~200 nm. Then, a Ti/Al/Ni/Au metal stack was deposited by the electron beam evaporation, and annealed at 850 °C for 30 s by the rapid thermal annealing, resulting in the formation of the source drain ohmic contact, as shown in Figure 1c. The source-drain spacing ($L_{SD}$) was 15 μm. Afterwards, as shown in Figure 1d, to reduce the gate leakage current, an oxygen plasma treatment was carried out with $O_2$ plasma asher. Figure 1e shows the gate Schottky contact with the Ni/Au metal deposition. The gate length ($L_G$), gate-source spacing ($L_{GS}$), and the gate-drain spacing ($L_{GD}$) were all 5 μm. Finally, the TFSI surface treatment was applied on the fabricated chip. Figure 2 shows the chemical structure schematic diagram of the TFSI. It was prepared in a glove box, and the diluents were 1, 2-dichloroethene (DCE) and 1, 2-dichlorobenzene (DCB). A TFSI solution with a solution concentration of 2 mg/mL was formed with 24 mg TFSI powder and 12 mL DCE. By using 0.5 mL of the 2 mg/mL TFSI solution and 4.5 mL DCB, a 0.2 mg/mL TFSI solution was produced. Then, the InAlN/GaN HEMTs were immersed in the 0.2 mg/mL TFSI solution for 20 s in ambient air conditions, then blow-dried with $N_2$. Both the device performances before and after TFSI treatment were measured using an Agilent B1500A semiconductor parameter analyzer (Agilent Technologies Inc., Santa Clara, CA, USA) to compare the effects of the TFSI on the InAlN/GaN HEMTs.

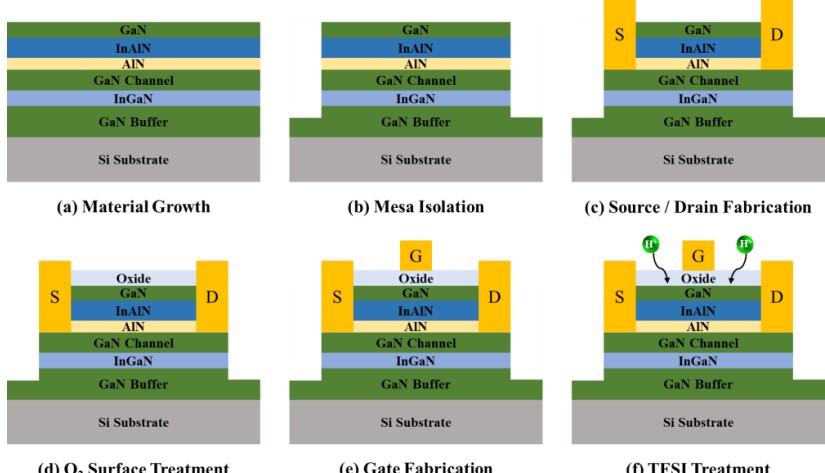

**Figure 1.** Schematic of the fabrication process for InAlN/GaN HEMT on Si substrate: (**a**) material growth, (**b**) mesa Isolation, (**c**) source/drain fabrication, (**d**) O$_2$ surface treatment, (**e**) gate fabrication, (**f**) TFSI surface treatment.

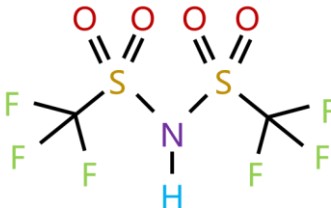

**Figure 2.** Chemical structure schematic diagram of the of bis(trifluoromethane) sulfonamide.

Figure 3a shows the $I_D$-$V_{DS}$ output characteristics of the InAlN/GaN HEMT before and after TFSI treatment. The on-resistance ($R_{on}$) is extracted in the linear region of the *I-V* curve under the gate-source voltage ($V_{GS}$) of 0 V and drain-source voltage ($V_{DS}$) between 0 V and 0.5 V. It was found that $R_{on}$ decreases from 8.74 Ω·mm (before TFSI) to 7.72 Ω·mm (after TFSI). The transfer characteristics at $V_{DS}$ = 10 V were measured and plotted in Figure 3b. The on-current increased from 0.36 A/mm (before TFSI) to 0.43 A/mm (after TFSI). The corresponding transfer characteristics in logscale are plotted in Figure 4a. The on/off ratio ($I_{ON}/I_{OFF}$), SS, and gate leakage current ($I_G$) were almost unchanged after TFSI surface treatment. Figure 4b exhibits the transconductance ($g_m$) at $V_{DS}$ = 10 V. With TFSI treatment, the peak $g_m$ increased from 118 mS/mm to 138 mS/mm.

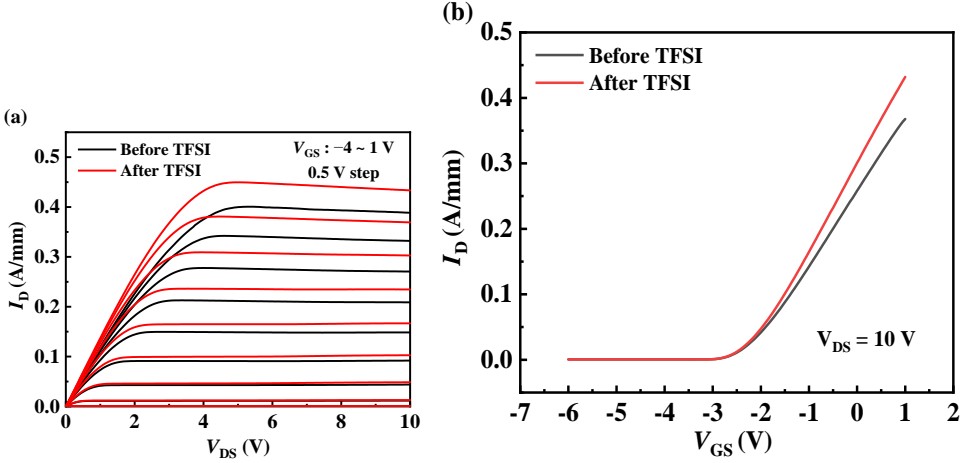

**Figure 3.** (**a**) *I-V* output characteristics of InAlN/GaN HEMT before and after TFSI treatment; (**b**) Transfer characteristics at $V_{DS}$ = 10 V of InAlN/GaN HEMT before and after TFSI treatment.

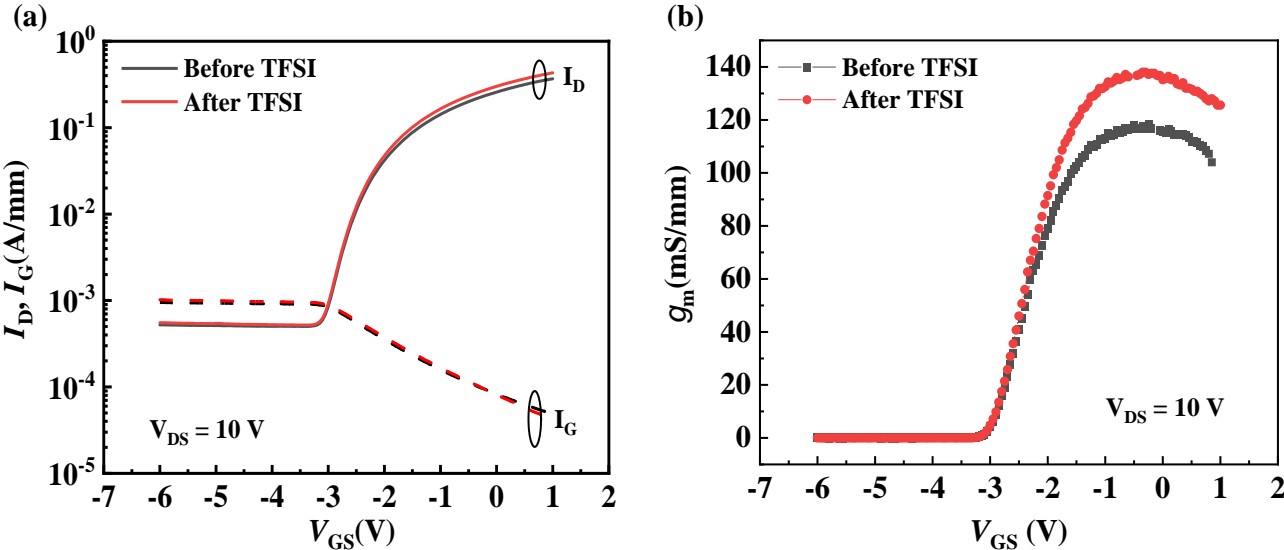

**Figure 4.** (**a**) Transfer characteristics in log-scale at $V_{DS}$ = 10 V of InAlN/GaN HEMT before and after TFSI treatment; (**b**) Transconductance at $V_{DS}$ = 10 V of InAlN/GaN HEMT before and after TFSI treatment.

To investigate the effect of TFSI treatment, the gate capacitance of both samples was measured, as shown in Figure 5a. By integrating *C-V* curves [14], the 2DEG electron density ($n_{2D}$) as a function of $V_{GS}$ was obtained, as shown in Figure 5b. It presents that the gate capacitance and the $n_{2D}$ are almost identical before and after TFSI surface treatment. The 2DEG electron mobility under the gate region is calculated as follows:

$$\mu_{n} = \frac{L_{G}}{en_{2D}W_{G}[V_{DS}/I_{DS} - (R_{D} + R_{S})]}, \tag{1}$$

$$R_{D} = \frac{L_{GD}}{en_{2D0}\mu_{n0}W_{G}}, \tag{2}$$

$$R_{S} = \frac{L_{GS}}{en_{2D0}\mu_{n0}W_{G}}. \tag{3}$$

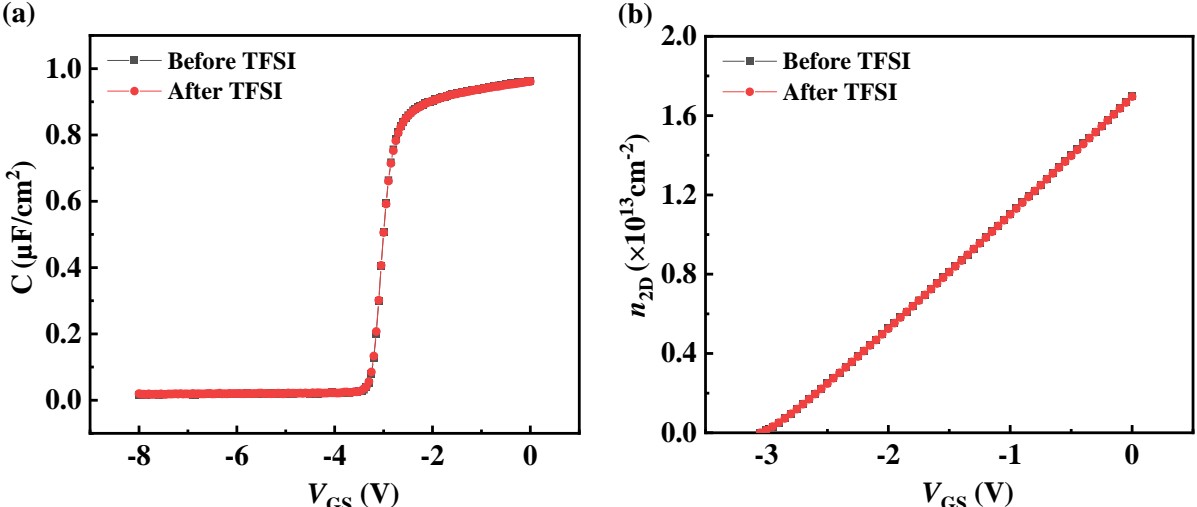

**Figure 5.** (**a**) Measured gate capacitance of InAlN/GaN HEMT before and after TFSI treatment; (**b**) Two-dimensional electron gas electron density ($n_{2D}$) of InAlN/GaN HEMT before and after TFSI treatment.

Here, $e$ is the electron charge, $V_{DS}$ and $I_{DS}$ are the drain-source voltage and current, respectively, $R_D$ and $R_S$ are the drain and source access resistance, respectively, and $n_{2D0}$ and $\mu_{n0}$ are the sheet density and electron mobility, respectively, at $V_{GS} = 0$ V. To reduce the influence of the lateral field from the drain voltage on the gate channel, the low drain voltage of 0.1 V was used for the electron mobility extraction. As shown in Figure 6, a significant improvement in the electron mobility was observed after TFSI surface treatment.

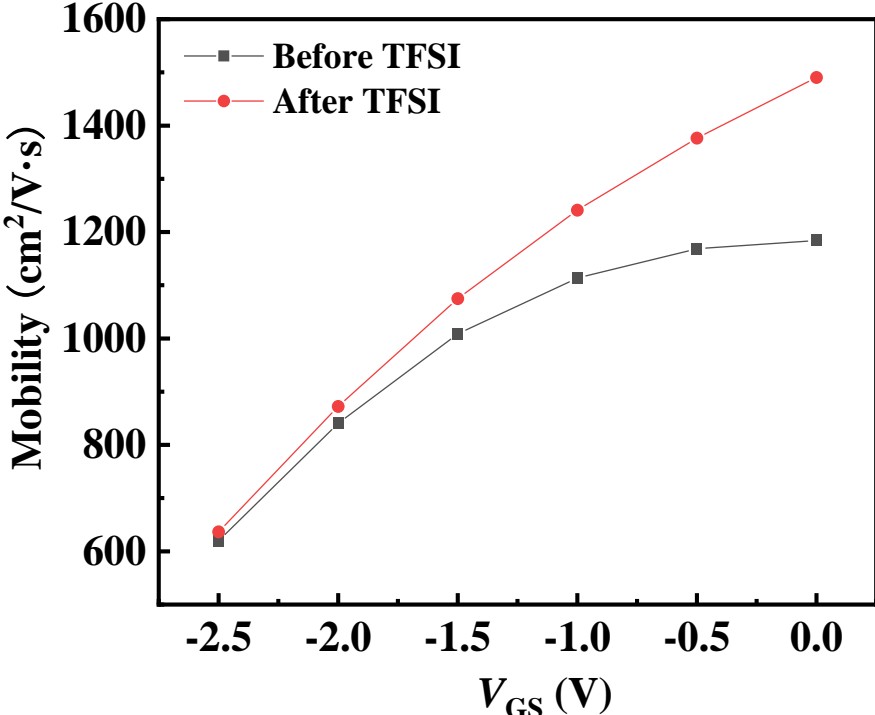

**Figure 6.** Electron mobility under the gate region of InAlN/GaN HEMT before and after TFSI treatment.

The electron mobility of the InAlN/GaN HEMTs is determined by the channel carrier scatterings, which includes polar optical phonon (POP), acoustic phonon (AP), interface roughness (IFR), dislocation (DIS), and polarization Coulomb field (PCF) scatterings [28–30]. The electron density (as shown in Figure 5b) of the InAlN/GaN HEMT before and after TFSI was almost identical, and the low channel field at $V_{DS} = 0.1$ V could not change the electron temperature. Therefore, POP, AP, IFR, and DIS scattering were unvaried after TFSI treatment [6]. Figure 7a,b show the schematic of the InAlN/GaN HEMT before and after TFSI surface treatment. Because of the large molecular structure, the TFSI molecule cannot diffuse through the oxide layer, which protects the GaN surface from becoming contaminated with larger TFSI molecules. However, the $H^+$ in TFSI could easily diffuse through the oxide layer [25,31]. Due to the spontaneous polarization of the InAlN barrier, there are polarization electric fields in the InAlN barrier, resulting in the negative polarization charges in the top and positive ones in the bottom of the InAlN barrier. The $H^+$ ions are near the top of the InAlN barrier and can neutralize the negative polarization charges in the top of the InAlN barrier, leading to the reduced strength of the electric field through the InAlN barrier. The decrease in the polarization electric field means a decrease in the polarization charges of the InAlN barrier under the access region, resulting in a decrease in the PCF scattering.

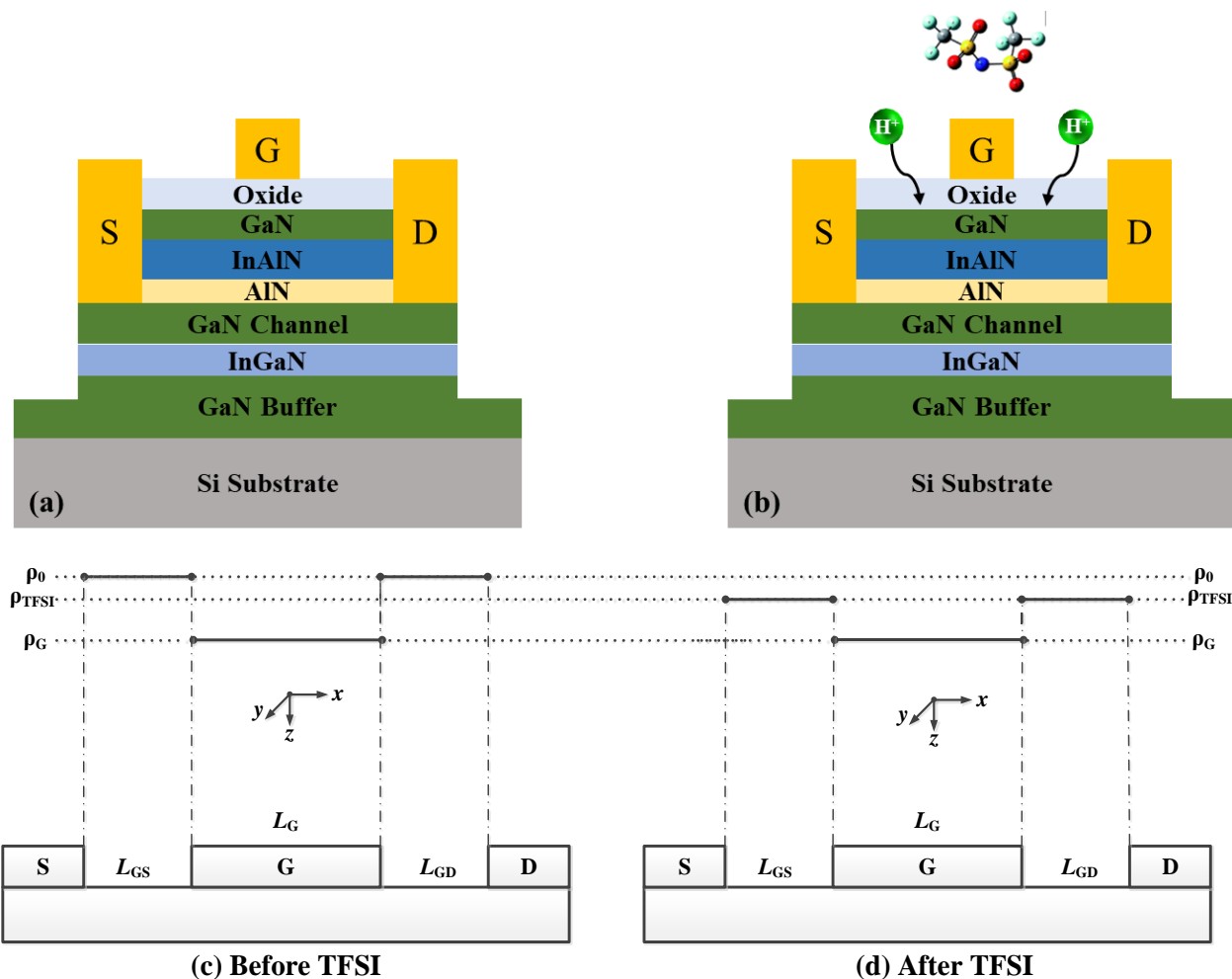

**Figure 7.** (**a**,**b**) Schematic of the InAlN/GaN HEMT before and after TFSI treatment; (**c**,**d**) Schematics of the polarization charge distribution of the InAlN barrier before and after TFSI treatment.

Figure 7c,d show the schematic of the polarization charge distribution of the InAlN barrier before and after TFSI treatment. Initially the polarization charges of InAlN barrier under the gate region and access regions are uniform. Due to the gate bias, the polarization electric field under the gate region will be changed with the inverse piezoelectric effect, resulting in the variation of the polarization charges. As shown in Figure 7c, the polarization charge distribution is not uniform, which leads to the PCF scattering. Before the treatment, the polarization charge density is labeled as $\rho_0$ and can be calculated by self-consistently solving Schrodinger's and Poisson's equations. Owing to the converse piezoelectric effect, the gate bias can change the number of the polarization charges under the gate region (labeled as $\rho_G$), resulting in the additional polarization charges $\Delta\sigma = \rho_G - \rho_0$ [32,33]. After TFSI treatment, the polarization charges under the access region are changed to $\rho_{\text{TFSI}}$ (shown in Figure 7d), leading to $\Delta\sigma_{\text{TFSI}} = \rho_G - \rho_{\text{TFSI}}$. Because $\rho_{\text{TFSI}}$ is smaller than $\rho_0$, $\Delta\sigma_{\text{TFSI}}$ is smaller than $\Delta\sigma$. The PCF scattering potential can be written as follows [34,35]:

$$V(x,y,z) = -\frac{e}{4\pi\varepsilon_s\varepsilon_0}\int_{-\frac{L_G}{2}}^{\frac{L_G}{2}} dx'\int_0^W \frac{\Delta\sigma}{\sqrt{(x-x')^2+(y-y')^2+z^2}}dy' \qquad (4)$$

$\varepsilon_s$ is the static dielectric constant of GaN, $\varepsilon_0$ is the vacuum dielectric permittivity. Here, the PCF scattering potential is dominated by the additional polarization charges $\Delta\sigma$. The larger the $\Delta\sigma$, the larger the PCF scattering potential. Here, $\Delta\sigma_{\text{TFSI}}$ is smaller than $\Delta\sigma$, therefore the device with TFSI treatment presents weaker PCF scattering.

The momentum relaxation rate due to PCF, POP, AP, DIS, and IFR scatterings ($\tau_{PCF}$, $\tau_{POP}$, $\tau_{AP}$, $\tau_{DIS}$, and $\tau_{IFR}$) can be calculated by the two-dimensional scattering theory [29,35]. The electron mobility limited by different scattering mechanisms then can be calculated by $\mu = e\tau/m^*$ and the total electron mobility under the gate region can be obtained with different scattering mechanisms by using $\mu_n = e(\tau_{POP} + \tau_{PCF} + \tau_{AP} + \tau_{DIS} + \tau_{IFR})/m^*$. Here $m^*$ is the electron effective mass. Figure 8 shows the calculated results of the electron mobility before and after treatment. Firstly, the extracted total electron mobility agrees well with the measured results, demonstrating the accuracy of extraction process. The mobility limited by POP, AP, DIS, and IFR scattering is almost identical before and after TFSI treatment. This means that the TFSI surface did not change the scattering strength of them. Meanwhile, the electron mobility limited by PCF scattering shows higher values after TFSI. This further confirms that the PCF scattering is suppressed with the treatment, and it dominates the improved electron mobility of the InAlN/GaN HEMTs.

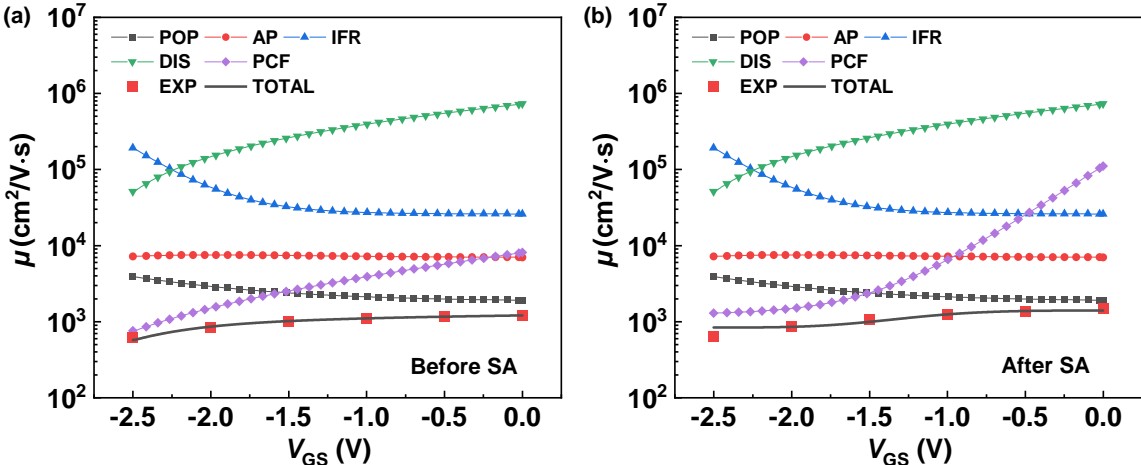

**Figure 8.** The calculated mobility as a function of gate-source voltage for PCF, POP, AP, IFR, and DIS scatterings as well as the total mobility (TOTAL) and the experimental mobility (EXP) at room temperature for the InAlN/GaN (**a**) Before TFSI surface treatment; (**b**) After TFSI surface treatment.

## 3. Conclusions

In summary, the effect of the TFSI surface treatment on InAlN/GaN HEMTs was demonstrated, and the improved electron mobility was investigated. The results indicated that the positive charge of H+ will decrease the polarization charges of the InAlN barrier under the access region, leading to reduced PCF scattering. This offers a possible way to improve the device performance of InAlN/GaN HEMTs for further application.

**Author Contributions:** Conceptualization, P.C. and Y.Z.; methodology, S.C., P.C.; investigation, S.C., P.C., M.X., Z.L., X.X., Y.Z. and J.H., writing—original draft preparation, S.C.; writing—review and editing, P.C. and Y.Z. All authors have read and agreed to the published version of the manuscript.

**Funding:** This research was funded in part by the Major Science and Technology Innovation Project of Shandong Province (grant no. 2022CXGC010103), in part by the Qilu Young Scholar of Shandong University, in part by the NASA International Space Station (grant nos. 80NSSC20M0142, 80NSSC22M0039 and 80NSSC22M0171), and in part by the Air Force Office of Scientific Research (grant nos. FA9550-21-1-0076 and FA9550-22-1-0126).

**Institutional Review Board Statement:** Not applicable.

**Informed Consent Statement:** Informed consent was obtained from all subjects involved in the study.

**Data Availability Statement:** Not applicable.

**Conflicts of Interest:** The authors declare no conflict of interest.

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
