# Peer review of "Improved Electrical Performance of InAlN/GaN High Electron Mobility Transistors with Post Bis(trifluoromethane) Sulfonamide Treatment"

_crystals, doi:10.3390/cryst12111521_

Round 1
Reviewer 1 Report
Improving electrical performance of heterojunctions is an important task for semiconductor science and technology. There are numerous scenarios to address this issue. One of them is surface treatment of heterojunctions to modify charge distribution. Therefore, the present work offering a way to increase electric conductance and performance of InAlN/GaN by its surface treatment by trifluoromethane) sulfonamide is on demand. Using a set of electron transport sensitive characterisation tools, the authors have found new results, such as a certain increase of two-dimensional electron mobility and concomitant reduction of on-state resistance and an increase of transconductance. The authors offer a mechanism that the charge of hydrogen ion decreases the polarization charges of the InAlN barrier leading to the reduced polarization Coulomb field scattering in InAlN/GaN structure.
The strength of the work: Combination of accurate sample preparation and a set of complementary characterisation tools, enabling quite convincing results relating to charge transport of the transistors. Quite high applied relevance of the work in terms of improvement of performance of transistors.
The weakness: The electric conductance and performance of the structures under investigation are dependent on the chemical and the electronic state of the elements involved. Therefore, the work would gain if the state of the surface were compared before and after treatment by the solution, for instance, by XPS.
In general, the work is scientifically sound, well arranged and conducted. In my view, it is suitable for publication in its present form, after careful check against misprints and punctuation, for instance, lines 48, 63, etc.
Author Response
We greatly appreciate your efficient, professional, and rapid processing of our manuscript entitled “Improved electrical performance of InAlN/GaN high electron mobility transistors with post bis(trifluoromethane) sulfonamide treatment” (Submission ID crystals-1972425) submitted to Crystals. The comments of reviewers are valuable and pertinent. After deliberated the comments from the reviewers, we have revised the manuscript accordingly. A point-by-point response to the previous reviewers' comments is described in the Response Letter to Reviewer 1.

Reviewer 2 Report
Good job.
I am just suggesting to check if "total" and "exp" curves "before" and "after" in Fig. 11 are not mixed up, as no difference can be seen.
And to improve the layout:
it is better when figures go after the text where them are mentioned;
to avoid break of chemical formula into two lines as on 85-86 lines.
Author Response
We greatly appreciate your efficient, professional, and rapid processing of our manuscript entitled “Improved electrical performance of InAlN/GaN high electron mobility transistors with post bis(trifluoromethane) sulfonamide treatment” (Submission ID crystals-1972425) submitted to Crystals. The comments of reviewers are valuable and pertinent. After deliberated the comments from the reviewers, we have revised the manuscript accordingly. A point-by-point response to the previous reviewers' comments is described in the Response Letter to Reviewer 2.

Reviewer 3 Report
In this manuscript titled ‘Improved electrical performance of InAlN/GaN high electron mobility transistors with post bis(trifluoromethane) sulfonamide treatment’, the authors have detailed investigated the influence of TFSI treatment of the InAlN/GaN HEMTs. This is a very interesting topic. The manuscript is well written and well organized including both experimental and calculation results. I recommend this manuscript should be published in Crystals after clarifying the following points.
1.Figure 3 to 9 need to be rearranged. Two or three sets of figures instead of 7 individual figures. The format of Figure 3 needs to be improved.
2. The sentence from Line 96 to 97 is quite confusing, which states that ‘As shown in Fig. 2, TFSI was prepared in a glove box.’ Also, figure 2 and its caption is identical to Fig.3 in reference 31. It would be good to have a different style of illustration.
3. Can the authors explain that why using TFSI but not using other organic superacid to provide H+? Anything special about TFSI compared with other acids?
Author Response
We greatly appreciate your efficient, professional, and rapid processing of our manuscript entitled “Improved electrical performance of InAlN/GaN high electron mobility transistors with post bis(trifluoromethane) sulfonamide treatment” (Submission ID crystals-1972425) submitted to Crystals. The comments of reviewers are valuable and pertinent. After deliberated the comments from the reviewers, we have revised the manuscript accordingly. A point-by-point response to the previous reviewers' comments is described in the Response Letter to Reviewer 3.
